# BRCA 1/2 Germline Mutation Predicts the Treatment Response of FOLFIRINOX with Pancreatic Ductal Adenocarcinoma in Korean Patients

**DOI:** 10.3390/cancers14010236

**Published:** 2022-01-04

**Authors:** Ji Hoon Park, Jung Hyun Jo, Sung Ill Jang, Moon Jae Chung, Jeong Youp Park, Seungmin Bang, Seung Woo Park, Si Young Song, Hee Seung Lee, Jae Hee Cho

**Affiliations:** 1Department of Internal Medicine, Division of Gastroenterology, Yonsei University College of Medicine, Seoul 03722, Korea; jihoon815@yuhs.ac (J.H.P.); JUNGHYUNJO83@yuhs.ac (J.H.J.); MJCHUNG@yuhs.ac (M.J.C.); sensass@yuhs.ac (J.Y.P.); bang7028@yuhs.ac (S.B.); swoopark@yuhs.ac (S.W.P.); sysong@yuhs.ac (S.Y.S.); 2Department of Internal Medicine, Gangnam Severance Hospital, Yonsei University College of Medicine, Seoul 06273, Korea; aerojsi@yuhs.ac

**Keywords:** breast cancer gene, *BRCA*, FOLFIRINOX, pancreatic ductal adenocarcinoma

## Abstract

**Simple Summary:**

In pancreatic ductal adenocarcinoma, FOLFIRINOX and nab-paclitaxel are recommended as first-line chemotherapy regimens. However, there are limited data to predict the efficacy of the FOLFIRINOX regimen in patient outcomes. Platinum-based chemotherapy is tolerable and responsible in patients with DNA damage repair gene mutations. However, data are still limited, and no Asian data are available yet. Here, we sought to investigate the proportion of germline *BRCA 1*/*2* mutations in patients with germline blood tests. Finally, we investigated the treatment response of FOLFIRINOX in patients with *BRCA 1*/*2* mutations. We found that the presence of germline *BRCA 1*/*2* mutations was associated with an improved overall response rate in pancreatic ductal adenocarcinoma patients treated with FOLFIRINOX. The high response rate in this analysis supports the preferential use of FOLFIRINOX therapy for patients harboring a *BRCA* germline mutation, and supports the need for early germline testing in order to select the best therapy.

**Abstract:**

We evaluated the proportion of *BRCA 1*/*2* germline mutations in Korean patients with sporadic pancreatic ductal adenocarcinoma (PDAC) and its effect on the chemotherapeutic response of FOLFIRINOX. This retrospective study included patients who were treated at two tertiary hospitals between 2012 and 2020, were pathologically confirmed to have PDAC, and had undergone targeted next-generation sequencing-based germline genetic testing. Sixty-six patients were included in the study (24 men; median age 57.5 years). In the germline test, *BRCA 1*/*2* pathogenic mutations were found in nine patients (9/66, 13%, *BRCA 1*, *n* = 3; *BRCA 2*, *n* = 5; and *BRCA 1*/*2*, *n* = 1). There was no significant difference in the baseline characteristics according to *BRCA* mutation positivity. Among patients who underwent FOLFIRINOX chemotherapy, patients with a *BRCA 1*/*2* mutation showed a higher overall response rate than those without a *BRCA 1*/*2* mutation (71.4% vs. 13.9%, *p* = 0.004). Patients with a germline *BRCA 1*/*2* mutation showed longer progression-free survival than those without a *BRCA 1*/*2* mutation, without a significant time difference (18 months vs. 10 months, *p* = 0.297). Patients with a *BRCA 1*/*2* mutation in the germline blood test had a higher response rate to FOLFIRINOX chemotherapy in PDAC. The high proportion of *BRCA 1*/*2* germline mutations and response rate supports the need for germline testing in order to predict better treatment response.

## 1. Introduction

Pancreatic ductal adenocarcinoma (PDAC) is expected to become the second leading cause of cancer-related deaths in the US before 2030 [1]. In the NCCN (National Comprehensive Cancer Network) guidelines, germline testing is recommended for patients with PDAC, using comprehensive gene panels for hereditary cancer syndromes [2]. The genes commonly associated with pathogenic germline alterations are *BRCA 1*/*2*, *ATM*, *PALB2*, *MLH1*, *MSH2*, *MSH6*, *PMS2*, *CDKN2A*, and *TP53* [3]. Among them, the frequency of detected *BRCA 1*/*2* (breast cancer susceptibility gene-1 and -2) is 4% to 7% [4,5]. The risk for pancreatic cancer is elevated two- to six-fold in these patients [6,7].

Recently, the POLO trial showed the benefit of poly ADP ribose polymerase (PARP) inhibitors in *BRCA* mutations. *BRCA* genes encode proteins involved in homologous recombination repair, and cells with mutations are sensitive to PARP inhibitors. However, there was no difference in overall survival between the PARP inhibitor and placebo groups (*p* = 0.68) [8]. Furthermore, in the real world, it is difficult for clinicians to change regimens in patients who are tolerant to FOLFIRINOX (oxaliplatin, irinotecan, folinic acid, and fluorouracil) chemotherapy.

In PDAC, FOLFIRINOX and nab-paclitaxel are recommended as first-line chemotherapy regimens. The guidelines recommend FOLFIRINOX in patients who are young and with better performance status (ECOG 0–1) [2]. However, there are limited data to predict the efficacy of the FOLFIRINOX regimen in patient outcomes [9]. Platinum-based chemotherapy is tolerable and responsible in patients with DNA damage repair gene mutations [10,11]. However, data are still limited, and no Asian data are available yet [12,13,14,15,16].

Here, we sought to investigate the proportion of germline *BRCA 1*/*2* mutations in patients with germline blood tests. Finally, we investigated the treatment response of FOLFIRINOX in patients with a *BRCA 1*/*2* mutation.

## 2. Materials and Methods

### 2.1. Study Population

This dual institutional retrospective analysis was performed on all patients diagnosed with PDAC who underwent a germline blood test between January 2012 and February 2020. We identified 66 patients who underwent a germline blood test. Of these, two patients were excluded from the study on account of insufficient clinical data (*n* = 2). One patient was diagnosed and treated at another hospital, and one patient died shortly after diagnosis due to deterioration of the condition. The remaining 64 patients were included in the analysis. This study was performed in accordance with the Declaration of Helsinki, as reflected by the institutional review board of Severance Hospital (approval number 4-2021-1151).

### 2.2. Variables

We evaluated patient characteristics, laboratory variables, tumor characteristics, progression-free survival (PFS), overall survival (OS), and overall response rate (ORR). Patient demographics and clinical characteristics, including age, sex, personal and family history of cancer, hypertension, diabetes mellitus, smoking history, body mass index (BMI), systemic chemotherapy, and response to treatment, were obtained from medical records and imaging studies. BMI, defined as body weight divided by the square of the height, was categorized following the guidelines of the World Health Organization (WHO 2000) (BMI < 18.5, underweight; 18.5–24.9, normal range; ≥25.0, overweight; and ≥30.0, obese). Tumor characteristics (location, extent, and number of metastatic organs) and laboratory characteristics (carbohydrate antigen [CA] 19-9) were also investigated.

The date of death and the date of the last follow-up were reviewed to estimate the OS and PFS. We observed both survival and follow-up data until 5 March 2021. OS was defined as the interval from the start of FOLFIRINOX until death. PFS was defined as the interval from the start of FOLFIRINOX to progressive disease (PD) or death. Patients who remained without death or PD were censored at the time of the last follow-up. Responses were determined using RECIST (response evaluation criteria in solid tumors) v1.1. ORR was defined as the percentage of patients who had a best response rating of complete response (CR) or partial response (PR) at any time point during treatment with chemotherapy. Patients without measurable disease at baseline were excluded from the ORR analysis.

### 2.3. DNA Extraction and Sequencing

Genomic DNA was extracted from peripheral blood using a QIAamp DNA Blood Mini Kit (Qiagen, Venlo, The Netherlands). The amount of input DNA was approximately 500 ng. DNA was fragmented into segments between 150 and 250 bp using the Bioruptor^®^ Pico sonication system (Diagenode, Liege, Belgium), end-repaired, and ligated to Illumina adapters (Illumina, San Diego, CA, USA) and indices. Sequencing libraries were hybridized with capture probes (Celemic, Seoul, Korea). The enriched DNA was then amplified, and clusters were generated and sequenced on a NextSeq 550 instrument (Illumina) with 2 × 151 bp reads [17]. Pathogenicity interpretations of the variants were performed according to the 2015 American College of Medical Genetics and Genomics guidelines by professional medical geneticists, using evidence from variant type assessments, population allele frequency, prediction algorithm results, and searches within databases such as ClinVar.

### 2.4. Statistical Analysis

The baseline demographics and characteristics of the patients were analyzed using descriptive statistics. The differences in baseline characteristics and ORR between *BRCA*-positive and *BRCA*-negative groups were analyzed using the chi-square test for categorical variables and the Student’s *t*-test for continuous variables. We estimated the median OS and PFS according to *BRCA* mutations using Kaplan–Meier curves and compared them using the log-rank test. A time-dependent Cox regression analysis was applied to estimate hazard ratios (HRs) with 95% confidence intervals (CIs) of pancreatic cancer mortality associated with *BRCA* mutations. Statistical significance was set at *p* < 0.05. All analyses were conducted using SPSS version 26.0 (SPSS, Chicago, IL, USA).

## 3. Results

### 3.1. Patients’ Characteristics and BRCA 1/2 Gene Mutations

A total of 66 PDAC patients underwent germline mutation analysis. Of all participants, three patients (4.5%) had a *BRCA1* mutation, five (7.6%) had a *BRCA2* mutation, and one patient (1.5%) had a *BRCA 1*/*2* mutation. None had germline *ATM* or *PALB2* mutations. Two patients had *KRAS* mutations, and one patient each had *TP53, CDK2NA, SMAD4,* and *MUTYH* mutations. Ten patients (15.2%) had *BRCA* variants of unknown significance (VUS). The specific *BRCA* mutations are listed in Appendix A. During the study period, somatic mutation tests were performed on 31 patients not included in this study. Two patients (2/31, 6.5%) had a *BRCA1* mutation, and two patients (2/31, 6.5%) had a *BRCA2* mutation. They were not included in this analysis. Of these 66 patients, 2 patients were excluded due to insufficient clinical data. Of these 64 patients, 7 patients had resectable PDAC, 20 patients had borderline resectable or locally advanced PDAC, and 37 patients had metastatic PDAC. Seven patients with resectable PDAC underwent curative intent resection (Figure 1). This study was performed in accordance with the Declaration of Helsinki, as reflected by the institutional review board of Severance Hospital (approval number 4-2021-1151).

The patient demographics and clinical characteristics are summarized in Table 1. The median age of the patients was 57.5 years (interquartile range, 48.0–66.8 years), and 37.5% were men. Overall, 29.7% (19/64) of the patients in our study also had a personal history of malignancy, including breast cancer (6/64, 9.4%), thyroid cancer (2/64, 3.1%), and ovarian cancer (1/64, 1.6%). A higher percentage of patients in the *BRCA*-positive group had a prior history of malignancy (6/9, 66.7% vs. 13/55, 23.6%, *p* = 0.016) and breast cancer (4/9, 44.4% vs. 2/55, 3.64%). Of the 20 patients who had a family history of cancer, 10 (22.7%) were of pancreatic cancer, 3 (6.8%) were of breast cancer, and 12 (27.3%) were of other malignancies. It was not possible to identify whether the family history of cancer was from a first-degree relative or not. The proportion of family history of any malignancy and number of metastatic sites was greater in the *BRCA*-positive group than in the *BRCA*-negative group; however, the difference was not significant (all, *p* > 0.05). The other variables showed no significant differences between the two groups. In the Kaplan–Meier survival analysis, there was no significant difference seen in OS between the *BRCA*-positive and *BRCA*-negative groups (*p* = 0.888) (Appendix A). The multivariable Cox regression model showed no significant improvement in OS in the presence of *BRCA* 1/2 mutations (HR, 0.128; 95% CI, 0.021–1.618) (Appendix A). The risk factors related to OS were tumor location (HR, 7.335; 95% CI, 2.030–26.503, *p* = 0.002), T stage (HR, 0.333; 95% CI, 0.115–0.963, *p* = 0.042), and M stage (HR, 7.661; 95% CI, 2.188–26.824, *p* = 0.001) in the multivariate analysis.

### 3.2. FOLFIRINOX Treatment and Overall Response Rate

In total, 47 patients of the study participants received FOLFIRINOX chemotherapy. Of these 47 patients, 4 had no response evaluation and were hence excluded from the ORR analysis. The patient demographics and clinical characteristics are summarized in Table 2. Of these 43 patients, 7 (16.3%) had a *BRCA* mutation. The median age of the patients was 51.0 years (interquartile range, 46.0–65.0 years), and 41.9% were male. Of the *BRCA*-positive group, 57.1% (4/7) had a history of prior malignancy, compared with 16.7% (6/36) of the *BRCA*-negative group, and there was a significant difference observed between the groups (*p* = 0.040). A history of breast cancer was reported in 42.9% (3/7) and 2% (2/36) of the *BRCA*-positive and *BRCA*-negative groups, respectively (*p* = 0.024). A median of 12.0 FOLFIRINOX cycles were administered to the *BRCA*-positive patients, and 9.0 cycles were administered to the *BRCA*-negative patients. FOLFIRINOX therapy was mostly administered in the first-line setting: 93.0% (40/43) in the first-line setting and 7.0% (3/43) in the second-line setting. The ORR, as defined by RECIST v1.1, was significantly higher in *BRCA*-positive patients than in *BRCA*-negative patients (5/7, 71.4% vs. 5/36, 13.9%; *p* = 0.004) (Table 3). For *BRCA*-positive patients, partial response and stable disease were observed in 71.4% (5/7) and 28.6% (2/7) of patients, respectively. None of the *BRCA*-positive patients showed a complete response. Of the 43 patients, 7 (16.3%) had a *BRCA* mutation and 5 (11.6%) had a *BRCA* mutation of unknown significance. The ORR was significantly higher in *BRCA*-positive patients, including those with mutations of unknown significance, than in *BRCA*-negative patients (7/12, 58.3% vs. 3/31, 9.7%; *p* = 0.002) (Appendix A). A subset analysis was performed to test the effect of gemcitabine/nab-paclitaxel on the ORR. Patients with a *BRCA 1*/*2* mutation did not show a significantly better response than those without a *BRCA 1*/*2* mutation (1/3, 33.3% vs. 0/17, 0.0%, *p* = 0.154).

### 3.3. Progression-Free Survival and Overall Survival

In our study, 71.2% (47/66) of the patients received FOLFIRINOX treatment. Of the 47 patients, 32 patients with locally advanced, metastatic, or recurrent PDAC were treated with palliative first-line FOLFIRINOX. These cases were included in the survival analyses. As a clinically relevant surrogate of the durability of FOLFIRINOX responses, we utilized PFS, defined as the date of first FOLFIRINOX chemotherapy administration to the date of clinical treatment failure. *BRCA*-positive patients had longer PFS than *BRCA*-negative patients. However, this association was not significant (*p* = 0.423). The median PFS was 18.0 months for *BRCA*-positive patients and 10.0 months for *BRCA*-negative patients. The OS for *BRCA*-positive and *BRCA*-negative patients did not reach the median (Figure 2).

## 4. Discussion

In this study, *BRCA 1*/*2* germline mutations predicted the treatment response of FOLFIRINOX in patients with PDAC. The PFS was longer in patients with a *BRCA 1*/*2* mutation than those with a wild type, even though the difference was not statistically significant. In this study, the rate of *BRCA 1*/*2* mutations was 13.6%. The data values were slightly higher than previous data (range 4–7%) in the general population [4,5]. The higher proportion of *BRCA 1*/*2* mutations may be due to the change in detection method with the adoption of next-generation sequencing. In addition, considering that a high proportion of patients were previously diagnosed with breast cancer in this study, the results are similar to those of previous studies. The prevalence of *BRCA 1*/*2* mutations in Asian patients with familial breast cancer and early-onset breast cancer was reported to be 2.8% to 31.8% [18]. Previous studies showed that *BRCA* gene mutations were associated with patients’ survival outcomes [12,13,19]. In this study, patients with *BRCA* gene mutations did not show different survival outcomes on account of the small number of patients.

The clinical significance and prognostic value of germline BRCA pathogenic mutations in tumors are well-known, but whether missense variants of uncertain significance (VUS) have clinical impact is not known. Variants in the gene were often classified as VUSs because of an insufficient understanding of the gene’s role. Variants can be reclassified from VUS to likely pathogenic, and further, to pathogenic. Phosphorylation of *BRCA 1*/*2* mutations plays an important role in their function as regulators of DNA repair, transcription, and cell cycles in response to DNA damage. Tram et al. suggested that VUS have the potential to interfere with the phosphorylation process via abolishing or creating phosphorylation sites on *BRCA 1*/*2* [20]. Hu et al. reported that germline VUS variant carriers had superior disease-free survival when compared with wild-type PDAC patients receiving adjuvant chemotherapy (16.5 months vs. 13.1 months, *p* = 0.007) [21]. Previous statistics indicate that between 10–20% of BRCA sequencing results are VUSs, and of these, more than 50% are missense mutations [22]. In this study, *BRCA 1*/*2* missense mutations (VUSs) were detected in 15.2% of our cohort (Appendix A). The ORR was significantly higher in BRCA-positive patients, including missense mutations of VUS, than in BRCA-negative patients (7/12, 58.3% vs. 3/31, 9.7%, *p* = 0.002). With the further accumulation of data in the future, VUS can be reclassified as pathogenic.

Previously, several studies reported on the proportion of *BRCA 1*/*2* mutations and their impact on patients with PDAC [9,12,13,16,19,23,24,25,26,27,28]. Golan et al. showed a difference in survival outcome for stage 3 or 4 PDAC patients with *BRCA 1*/*2* mutations in platinum-based chemotherapy (22 months vs. 9 months, *p* = 0.039) [19]. Wattenberg et al. reported on the treatment response of platinum-based chemotherapy in PDAC patients with *BRCA 1*/*2* mutations (58% vs. 21%, *p* = 0.002) [12]. In the present study, patients who received FOLFIRINOX chemotherapy showed a better treatment response in *BRCA*-positive patients compared to *BRCA*-negative patients. However, patients who received nab-paclitaxel chemotherapy did not show any difference in treatment response, irrespective of *BRCA* mutations.

Recently, several studies attempted to identify patients who benefit from palliative first-line FOLFIRINOX chemotherapy. Transcriptomic analysis showed that the basal type showed a better treatment response to FOLFIRINOX chemotherapy. The immunohistochemistry stained marker KRT81 may be a predictive marker to identify patients in the clinical field [29]. Circulating blood markers, such as ctDNA and exosomes, were also suggested as predictors for FOLFIRINOX response [30]. In other studies, protein markers, CES2 expression, and female gender predicted the response to FOLFIRINOX in PDAC [31,32]. The ideal predictor is a non-invasive clinically feasible tool during patient treatment. In this study, *BRCA 1*/*2* was a predictor of the response to FOLFIRINOX. However, the proportion of *BRCA* cases was very low in patients with PDAC. Several clinical trials are currently ongoing to identify better blood germline biomarkers (ClinicalTrials.gov NCT04289961; NCT04143152).

Despite the efficacy of *BRCA* on treatment response in patients, the present study did not show survival benefits in patients who underwent FOLFIRINOX. Regardless of how good a prognostic or therapeutic predictive marker may be, it cannot outperform clinical parameters, such as cancer stage, age, sex, and metastasis, on their prognosis. Germline mutations can be used to predict FOLFIRINOX treatment response; however, they are still limited in predicting patient prognosis. A previous study by Sehdev et al. and Golan et al. also showed a significant difference in the prognosis of BRCA-positive patients who received platinum-based chemotherapy [16,24]. 

Our study has strengths. This is the first report of the ORR in numerous patients with *BRCA* 1/2 mutations following the use of FOLFIRINOX in Asia. In a previous study, less was known about the prevalence and treatment outcomes of FOLFIRINOX involving *BRCA* 1/2 mutations in Asia [14]. The high ORR of 71.4% with FOLFIRINOX therapy in *BRCA*-positive patients suggests that platinum therapy may be particularly desirable for this subset of patients in clinical scenarios marked by high disease burden and symptomatic disease, and for patients with PDAC. This study may help guide treatment decisions for patients with PDAC.

This study has several limitations. First, this is a retrospective study. Although we adjusted several factors via multivariate analysis, selection and/or information bias could remain. The lack of statistically significant differences in both OS and PFS in this study population may be attributed to the limited number of patients enrolled in the study: only 32 patients in our study were treated with palliative, first-line FOLFIRINOX. Second, although we found no significant difference in the proportion of males between groups, there were fewer males in the *BRCA*-positive group (2/9, 22.2% vs. 22/55, 40.0%, *p* = 0.464) [32]. In this study, relatively young patients were enrolled compared to previous studies (median 57.5 years) [12,19,33]. Both findings could plausibly skew bias toward the null hypothesis. Third, we were unable to control for mortality comorbidities that might have affected our results. However, since FOLFIRINOX is indicated for relatively healthier PDAC patients with good performance status, we do not think that the difference in comorbidities is the only explanation for our results.

## 5. Conclusions

We found that the presence of germline *BRCA 1*/*2* mutations is associated with an improved ORR in PDAC patients treated with FOLFIRINOX. These results validate the association of germline *BRCA 1*/*2* mutations with platinum sensitivity, as reported by other results in patients with PDAC. Notably, the high response rate in this analysis supports the preferential use of FOLFIRINOX therapy for patients with PDAC harboring a *BRCA* germline mutation, and supports the need for early germline testing in order to select the best therapy. Further prospective studies are needed to refine the treatment paradigms for this important subset of patients with PDAC.

## Figures and Tables

**Figure 1 cancers-14-00236-f001:**
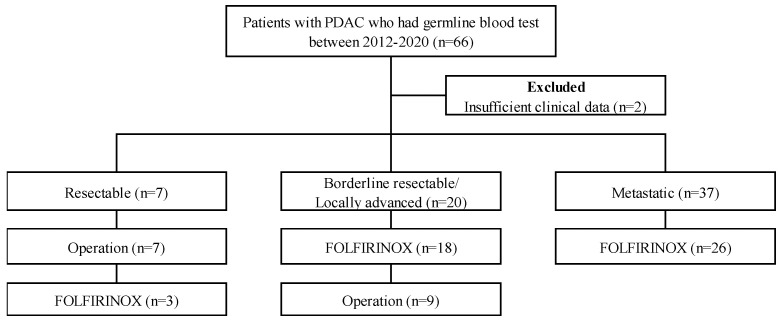
Selection of study population with PDAC, pancreatic ductal adenocarcinoma; FOLFIRINOX: oxaliplatin, irinotecan, folinic acid, and fluorouracil.

**Figure 2 cancers-14-00236-f002:**
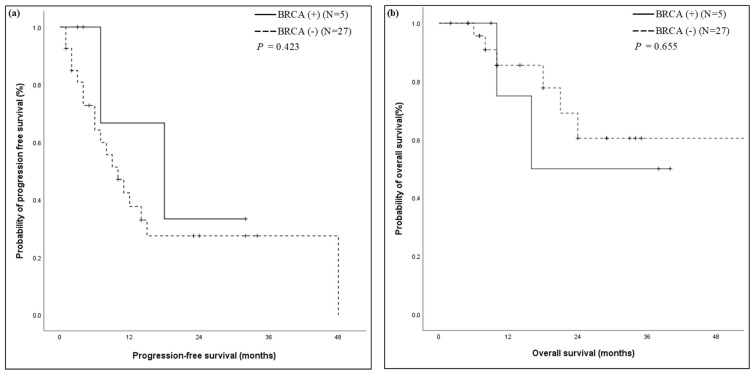
Kaplan–Meier curve in patients with first-line FOLFIRINOX stratified by the presence of a germline *BRCA* gene mutation. (**a**) The PFS was 18 months (95% confidence interval 0.4–35.6) in the *BRCA*-positive group, as compared to 10 months (95% confidence interval 5.5–14.5) in the *BRCA*-negative group (*p* = 0.423); (**b**) Kaplan–Meier survival curves for overall survival of the patients were not reached. *FOLFIRINOX: oxaliplatin, irinotecan, folinic acid, and fluorouracil; BRCA: breast cancer susceptibility gene*.

**Table 1 cancers-14-00236-t001:** Baseline characteristics of patients who had germline genetic blood tests.

Variables		Total (*n* = 64)		*BRCA* Mutation (+)	*BRCA* Mutation (–)	*p* Value
				(*n* = 9)	(*n* = 55)	
Age at diagnosis (year)		57.5 (48.0–66.8)		50.0 (47.0–60.0)	59.0 (48.0–71.0)	0.122
Male		24 (37.5%)		2 (22.2%)	22 (40.0%)	0.464
History of prior malignancy, *n* (%)						

Yes		19 (29.7%)		6 (66.7%)	13 (23.6%)	0.016
Breast		6 (9.4%)		4 (44.4%)	2 (3.64%)	0.014
Family history of any malignancy, *n* (%)						

Yes		29 (45.3%)		6 (66.7%)	23 (41.8%)	0.279
Pancreas		14 (21.9%)		2 (22.2%)	12 (21.8%)	1.000
Breast		3 (4.7%)		1 (11.1%)	2 (3.6%)	1.000
Tobacco use (%)						
Yes (past or current)		16 (36.4%)		3 (33.3%)	19 (34.5%)	1.000
BMI (kg/m^2^)						
≥25.0		9 (14.1%)		1 (11.1%)	8 (14.5%)	1.000
Diabetes Mellitus		18 (28.1%)		1 (11.1%)	17 (30.9%)	0.425
Hypertension		21 (32.8%)		2 (22.2%)	19 (34.5%)	0.706
CA 19-9 (U/mL)						
Elevated (>34.0U/mL)		51 (79.7%)		8 (88.9%)	43 (78.2%)	0.672
Pathology						
Well-differentiated		2 (3.1%)		0 (0.0%)	2 (3.6%)	
Moderately differentiated		24 (37.5%)		7 (77.8%)	17 (30.9%)	
Poorly differentiated		7 (10.9%)		2 (22.2%)	5 (9.1%)	
Clinical T stage						
T1/2		25 (39.1%)		3 (33.3%)	22 (40.0%)	1.000
T3/4		39 (60.9%)		6 (66.7%)	33 (60.0%)	
Clinical *n* stage						
N0		26 (40.6%)		5 (55.6%)	21 (38.2%)	0.467
Location of primary tumor						
Head		32 (50.0%)		4 (44.4%)	28 (50.9%)	1.000
Metastasis site						
Liver		22 (34.4%)		6 (66.7%)	16 (29.1%)	0.053
Peritoneum		13 (20.3%)		0 (0.0%)	13 (23.6%)	0.185
Distant LN		9 (14.1%)		1 (11.1%)	8 (14.5%)	1.000
Number of metastasis site						
0 site		33 (51.6%)		3 (33.3%)	30 (54.5%)	0.296
1 or more sites		31 (48.4%)		6 (66.7%)	25 (45.5%)	

Data are in n (%) or median (IQR). BRCA: breast cancer susceptibility gene; BMI: body mass index; CA: carbohydrate antigen; LN: lymph node; IQR: interquartile range.

**Table 2 cancers-14-00236-t002:** Baseline characteristics of the patients who received FOLFIRINOX treatment and had response evaluation results.

Variables		Total (*n* = 43)		*BRCA* Mutation (+)	*BRCA* Mutation (–)	*p* Value
				(*n* = 7)	(*n* = 36)	
Age at diagnosis (year)		51.0 (46.0–65.0)		49.0 (46.0–56.0)	51.5 (44.5–65.8)	0.508
Male		18 (41.9%)		2 (28.6%)	16 (44.4%)	0.680
History of prior malignancy, *n* (%)						

Yes		10 (23.3%)		4 (57.1%)	6 (16.7%)	0.040
Breast		5 (11.6%)		3 (42.9%)	2 (5.6%)	0.024
Family history of any malignancy, *n* (%)						

Yes		22 (51.2%)		6 (85.7%)	16 (44.4%)	0.095
Pancreas		10 (23.3%)		2 (28.6%)	8 (22.2%)	0.656
Breast		3 (7.0%)		1 (14.3%)	2 (5.6%)	0.421
Tobacco use (%)						
Yes (past or current)		15 (34.9%)		3 (42.9%)	12 (33.3%)	0.680
BMI (kg/m^2^)						
≥25.0		9 (20.9%)		1 (14.3%)	5 (13.9%)	1.000
Diabetes Mellitus		9 (20.9%)		0 (0.0%)	9 (25.0%)	0.314
Hypertension		11 (25.6%)		2 (28.6%)	9 (25.0%)	1.000
CA 19-9 (U/mL)						
Elevated (>34.0 U/mL)		36 (83.7%)		6 (85.7%)	30 (83.3%)	1.000
Pathology						
Well-differentiated		1 (2.3%)		0 (0.0%)	1 (2.8%)	
Moderately differentiated		16 (37.2%)		5 (71.4%)	11 (30.6%)	
Poorly differentiated		5 (11.6%)		2 (28.6%)	3 (8.3%)	
Clinical T stage						
T1/2		16 (37.2%)		2 (28.6%)	14 (38.9%)	0.695
T3/4		27 (62.8%)		5 (71.4%)	22 (61.1%)	
Clinical *n* stage						
N0		16 (37.2%)		3 (42.9%)	13 (36.1%)	1.000
Location of primary tumor						
Head		22 (51.2%)		3 (42.9%)	19 (52.8%)	0.698
Metastasis site						
Liver		17 (39.5%)		5 (71.4%)	12 (33.3%)	0.093
Peritoneum		8 (18.6%)		0 (0.0%)	8 (22.2%)	0.315
Distant LN		5 (11.6%)		1 (14.3%)	4 (11.1%)	1.000
Number of metastasis site						
0 site		19 (44.2%)		2 (28.6%)	17 (47.2%)	0.243
1 or more sites		24 (55.8%)		5 (71.4%)	19 (52.8%)	

Data are in n (%) or median (IQR). BRCA: breast cancer susceptibility gene; BMI: body mass index; CA: carbohydrate antigen; LN: lymph node; IQR: interquartile range.

**Table 3 cancers-14-00236-t003:** Overall response rate to FOLFIRINOX in patients with a germline *BRCA* 1/2 mutation was significantly higher than patients without the mutation.

Outcome		*BRCA* Mutation (+)	*BRCA* Mutation (-)	*p* Value
		(*n* = 7)	(*n* = 36)	
Overall response rate, *n* (%)		5 (71.4%)	5 (13.9%)	0.004
Complete response		0	0	
Partial response		5 (71.4%)	5 (13.9%)	
Stable disease		2 (28.6%)	26 (72.2%)	
Progressive disease		0	5 (13.9%)	
Line of FOLFIRINOX therapy				
First		6 (85.7%)	34 (94.4%)	0.421
Second		1 (14.3%)	2 (5.6%)	

Response criteria according to RECIST (response evaluation criteria in solid tumours) v1.1. FOLFIRINOX: oxaliplatin, irinotecan, folinic acid, and fluorouracil; BRCA: breast cancer susceptibility gene.

## Data Availability

The data presented in this study are available on request from the corresponding author.

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
