# Peer review of "BRCA 1/2 Germline Mutation Predicts the Treatment Response of FOLFIRINOX with Pancreatic Ductal Adenocarcinoma in Korean Patients"

_cancers, 2022, doi:10.3390/cancers14010236_

Round 1

Reviewer 1 Report

This is a retrospective review of therapy of 66 patients with pancreatic cancer, of whom 9 had germline BRCA mutations. The principal problem with the manuscript is that after dividing the patients into subgroups, the numbers of BRCA+ patients available for evaluation is vanishingly small and this valid statistics cannot be carried out. As higher sensitivity to platinum agents is known, the study dose not enough useful information on the subject.

Other major points: somatic mutations were not carried out and other important mutationts (MSI) were also not tested.

Author Response

This is a retrospective review of therapy of 66 patients with pancreatic cancer, of whom 9 had germline BRCA mutations. The principal problem with the manuscript is that after dividing the patients into subgroups, the numbers of BRCA+ patients available for evaluation is vanishingly small and this valid statistics cannot be carried out. As higher sensitivity to platinum agents is known, the study dose not enough useful information on the subject.

Other major points: somatic mutations were not carried out and other important mutationts (MSI) were also not tested.

Reply:

We appreciate the reviewer’s keen comment. We fully agree that our study has limitation because sample size as seen in our experimental result is small. Further larger studies are needed to validate the association of germline BRCA mutations with platinum sensitivity. Pancreatic cancer is the eighth most common cancer (with an annual incidence of 7,032 in 2017) in South Korea. Because genetic testing for pancreatic cancer is not covered by insurance in South Korea, small number of patients were enrolled in this study and relatively young patients or patients who have a family history of cancer were enrolled compared to previous studies.

We agree that higher sensitivity to platinum agents is known for BRCA (+) patients. However, in a previous study, less was known about the prevalence and treatment outcome of FOLFIRINOX of BRCA 1/2 mutation in South Korea. Despite the recommendation of NCCN guideline [1], since germline testing is not covered by insurance for pancreatic cancer patients in South Korea, this study could help encourage germline testing.

Previous studies by Sehdev et al. and Palacio et al. showed a significant difference in the prognosis of somatic and germline BRCA-positive patients who received platinum-based chemotherapy [2,3]. Golan et al. and Wattenberg et al. reported the association of BRCA mutations and platinum-agents in PDAC patients with germline BRCA mutations [4,5]. Since germline testing is recommended for patients with PDAC in the NCCN guidelines, we analyzed germline BRCA mutations in the present study. According to reviewer’s comments, we further reviewed the patient data. Of 66 patients, microsatellite instability (MSI) test on tumor tissue was performed in 13 patients. Only 2 patients (2/13, 15.4%) had a MSI-H. Of 66 patients, no one received somatic mutation test in further review. During the study period, somatic mutation tests were performed in other 31 patients who were not included in this study. Two patients (2/31, 6.5%) had BRCA1 mutation and 2 patients (2/31, 6.5%) had BRCA2 mutation. We revised the manuscript and added this information in the result section.

In Result, page 3, line 132-136

After :

Ten patients (15.2%) had a BRCA variants of unknown significance. The specific BRCA mutations are listed in Supplementary Table 1. Of 66 patients, microsatellite instability test on tumor tissue was performed in 13 patients. Only 2 patients (2/13, 15.4%) had a MSI-H. During the study period, somatic mutation tests were performed in 31 patients not included in this study. 2 patients (2/31, 6.5%) had BRCA1 mutation and 2 patients (2/31, 6.5%) had BRCA2 mutation. They were not included in this analysis. Of these 66 patients, two patients were excluded due to insufficient clinical data.

Reviewer 2 Report

In the present manuscript, JH Park and colleagues report, in a unique Korean pancreatic cancer patient cohort, that patients carrying germline BRCA1/2 mutations showed higher objective response rate to platinum-based FOLFIRINOX therapy. The authors also show that FOLFIRINOX chemotherapy administration prolonged (non-significant) progression-free survival of BRCA-mutated patients. They finally conclude that these results support the need for early germline testing for pancreatic cancer patients and the preferential use of FOLFIRINOX therapy for BRCA patients. Overall, the study was well-conducted and reveal interesting data on BRCA-mutated patients in a unique Asian pancreatic ductal adenocarcinoma patient cohort.

The manuscript is a valuable contribution to the already existing studies showing that germline BRCA mutations create a certain vulnerability to platinum agents. However, even if this topic is currently of high interest and the study/approach valuable, it needs to be revised and improved.

  1. In general, to improve its readability, the manuscript requires an extensive revision and editing of the English language. As examples, the following sentences are really unclear or somehow misleading:

- “Furthermore, in the real world, clinicians seldom find it difficult to change chemotherapy regimens in patients who are tolerant to FOLFIRINOX (oxaliplatin, irinotecan, folinic acid and fluorouracil) chemotherapy”.

- “Several studies show that platinum-based chemotherapy is responsible for patients who undergo the FOLFIRNOX regimen”.

  1. In their study, the authors used a cohort of pancreatic cancer patients, retrospectively. Can the authors extend their study to other homologous recombination gene-mutated patients e.g. ATM, PALB2 mutation carriers? This will dramatically improve the importance and usefulness of their study, especially as they have access to a unique Asian (Korean) cohort.

  1. The authors should describe which tool or algorithm was used to assess pathogenicity of identified BRCA1/2 mutations.

  1. BRCA1/2 mutations of unknown significance were identified in the patient cohort. Could the authors also observe higher ORR after FOLFIRINOX treatment in these patients?

  1. In their study, the authors identified 66 patients who underwent a germline blood test. To possibly highlight the possible improvement that could be made with a more global testing strategy, what was the percentage of tested patients in their cohort? The authors finally conclude that their data support the need for early germline testing of pancreatic cancer patients. The NCCN guidelines also recommend germline testing for any confirmed PDAC patient. Can the authors explicit their opinion on that aspect? According to their data, would they recommend testing any confirmed PDAC patient?

  1. The second last paragraph of the discussion section describing the strengths of the study (page 9, line 246-253) requires rephrasing to improve its readability. Could the authors also further develop the “Second, we used a high quality, well-annotated clinical database” part by describing it better?

  1. Is there an objective explanation why their PDAC cohort is younger than other cohorts already described, as mentioned in the discussion section (page 9, line 261-262)?

  1. Can the authors shortly mention, at least in the Materials and Methods section, how the family history of cancer was evaluated: reported cancer types, first-degree relatives?

  1. The authors must describe the meaning of each mentioned abbreviation for each figure and table, in the corresponding legends. The same applies for the manuscript as some abbreviations are not described e.g. NCCN (page 2, line 45). Moreover, to improve the readability of the figure and table legends, can the authors add a comma between the abbreviation and their meaning and the semi-colon between terms, when they list them?

  1. The authors must correct the styling of gene names in order to always respect the gene nomenclature recommendations provided by the HGNC (italics for mRNAs and genes, capitals for proteins, etc.) in the manuscript and in tables.

  1. The equal contribution of first coauthors looks weirdly mentioned in the first part of manuscript: “These authors contributed equally as second coauthors” (page 1, line 13).

  1. The reviewer recommends a careful double checking of the spelling mistakes and typos, as numerous ones can be found across the manuscript, e.g. “FOLFIRNOX” (page 1, line 20 and page 2, line 65), “polyp ADP ribose polymerase” (page 2, line 52), “varient” (Supplementary Table 1).

  1. Finally, I would kindly recommend the authors to reconsider their article title, as it could be easily shortened and it mentioned two times “in patients” and “in Korean patients”.

Author Response

In the present manuscript, JH Park and colleagues report, in a unique Korean pancreatic cancer patient cohort, that patients carrying germline BRCA1/2 mutations showed higher objective response rate to platinum-based FOLFIRINOX therapy. The authors also show that FOLFIRINOX chemotherapy administration prolonged (non-significant) progression-free survival of BRCA-mutated patients. They finally conclude that these results support the need for early germline testing for pancreatic cancer patients and the preferential use of FOLFIRINOX therapy for BRCA patients. Overall, the study was well-conducted and reveal interesting data on BRCA-mutated patients in a unique Asian pancreatic ductal adenocarcinoma patient cohort.

The manuscript is a valuable contribution to the already existing studies showing that germline BRCA mutations create a certain vulnerability to platinum agents. However, even if this topic is currently of high interest and the study/approach valuable, it needs to be revised and improved.

  1. In general, to improve its readability, the manuscript requires an extensive revision and editing of the English language. As examples, the following sentences are really unclear or somehow misleading:

- “Furthermore, in the real world, clinicians seldom find it difficult to change chemotherapy regimens in patients who are tolerant to FOLFIRINOX (oxaliplatin, irinotecan, folinic acid and fluorouracil) chemotherapy”.

- “Several studies show that platinum-based chemotherapy is responsible for patients who undergo the FOLFIRNOX regimen”.

Reply: Thank you for your important comment. According to reviewer’s comments, we performed an extensive revision by specialized english editing company (Editage) and native english speaker, respectively. We added the editing certification. We also modified below sentences.

In Introduction, page 2, line 55-57

After :

Furthermore, in the real world, it is difficult for clinicians to change regimens in patients who are tolerant to FOLFIRINOX (oxaliplatin, irinotecan, folinic acid and fluorouracil) chemotherapy.

In Introduction, page 2, line 64-65

After :

This sentence has been deleted because it overlaps with the previous sentence.

  1. In their study, the authors used a cohort of pancreatic cancer patients, retrospectively. Can the authors extend their study to other homologous recombination gene-mutated patients e.g. ATM, PALB2 mutation carriers? This will dramatically improve the importance and usefulness of their study, especially as they have access to a unique Asian (Korean) cohort.

Reply: Thank you for your valuable comment. Wattenberg et al. reported the treatment response of platinum-based chemotherapy in PDAC patients with BRCA 1/2 mutation (58% vs. 21%, P=0.002) [4]. In this study, 26 patients were identified with the mutational breakdown: BRCA1 (n=5), BRCA2 (n=17) and PALB2 (n=4). Of Our 66 patients, none had germline ATM or PALB2 mutations. 2 patients had KRAS mutation and 1 patient each had TP53, CDK2NA, SMAD4, MUTYH mutation. We added this information in the result section.

In Result, page 3, line 129-130

After :

Of all participants, 3 patients (4.5%) had a BRCA1 mutation, 5 (7.6%) had a BRCA2 mutation and 1 patient (1.5%) had a BRCA1/2 mutation. None had germline ATM or PALB2 mutation. 2 patients had KRAS mutation and 1 patient each had TP53, CDK2NA, SMAD4, MUTYH mutation. 10 patients (15.2%) had a BRCA variants of unknown significance.

  1. The authors should describe which tool or algorithm was used to assess pathogenicity of identified BRCA1/2 mutations.

Reply: Thank you for your important comment. We added the detailed information about pathogenicity interpretation of the variants in Materials and Methods section.

In Materials and Methods, page 3, line 110-113

After :

Enriched DNA was then amplified, and clusters were generated and sequenced on a NextSeq 550 instrument (Illumina) with 2 x 151bp reads [17]. Pathogenicity interpretation of the variants were performed according to 2015 American College of Medical Genetics and Genomics guideline by professional medical geneticists, using evidences of variant type assessment, population allele frequency, prediction algorithm results, and database search such as ClinVar.

  1. BRCA1/2 mutations of unknown significance were identified in the patient cohort. Could the authors also observe higher ORR after FOLFIRINOX treatment in these patients?

Reply: Thank you for your valuable comment. In our study, 10 patients (15.2%) had a BRCA variants of unknown significance. We agree with your suggestion. We analyzed patients with BRCA1/2 mutations of unknown significance which you recommended. We inserted supplementary table 3.

In Result, page 6, line 186-189

After :

For BRCA-positive patients, partial response and stable disease were observed in 71.4% (5/7) and 28.6% (2/7) of patients, respectively. None of the BRCA-positive patients showed a complete response. Of the 43 patients, 7 (16.3%) had a BRCA mutation and 5 (11.6%) had a BRCA mutations of unknown significance. The ORR was significantly higher in BRCA-positive patients including mutations of unknown significance than in BRCA-negative patients (7/12, 58.3% vs. 3/31, 9.7%; P = 0.002) (Supplementary Table 3).

Supplementary Table 3. Overall response rate to FOLFIRINOX in patients with germline BRCA 1/2 mutation including variant of unknown significance was significantly higher than patients without mutation

Outcome

BRCA mutation (+)

BRCA mutation (-)

P value

(n=12)

(n=31)

Overall response rate, N(%)

7 (58.3%)

3 (9.7%)

0.002

 Complete response

0

0

 Partial response

7 (58.3%)

3 (9.7%)

 Stable disease

4 (33.3%)

24 (77.4%)

 Progressive disease

1 (8.3%)

4 (12.9%)

Line of FOLFIRINOX therapy

 First

10 (83.3%)

30 (96.8%)

0.184

 Second

2 (16.7%)

1 (3.2%)

Response criteria according to RECIST (response evaluation criteria in solid tumours) v1.1
FOLFIRINOX, oxaliplatin, irinotecan, folinic acid and fluorouracil; BRCA breast cancer susceptibility gene

  1. In their study, the authors identified 66 patients who underwent a germline blood test. To possibly highlight the possible improvement that could be made with a more global testing strategy, what was the percentage of tested patients in their cohort? The authors finally conclude that their data support the need for early germline testing of pancreatic cancer patients. The NCCN guidelines also recommend germline testing for any confirmed PDAC patient. Can the authors explicit their opinion on that aspect? According to their data, would they recommend testing any confirmed PDAC patient?

Reply: Thank you for your important comment. Pancreatic cancer is the eighth most common cancer (with an annual incidence of 7,032 in 2017) in South Korea. We have 2,832 patients in Pancreatic Cancer Cohort Registry of Severance Hospital in Seoul, Korea. Of the 2,832 patients, 66 patients available for evaluation is small. Because germline testing for pancreatic cancer is not covered by insurance in South Korea, we were unable to perform germline testing on all PDAC patients. Relatively young patients or patients who have a family history of cancer were enrolled. Recently, the rate of germline testing is increasing. We recommend germline test for any confirmed PDAC patients. As in our study, germline testing can help with regimen selection. Furthermore, patients may have the opportunity to use a drug like poly ADP ribose polymerase (PARP) inhibitor (POLO trial). Also, we may be able to do further studies that will help with PDAC patients in the future.

  1. The second last paragraph of the discussion section describing the strengths of the study (page 9, line 246-253) requires rephrasing to improve its readability. Could the authors also further develop the “Second, we used a high quality, well-annotated clinical database” part by describing it better?

Reply: Thank you for your careful comment. We agree with your opinion. We deleted the ambiguous sentence and we corrected the paragraph as follows.

In Discussion, page 9, line271-272

After :

Our study had strength. This was the first report of ORR in numerous patients with BRCA 1/2 mutation following the use of FOLFIRINOX in Asia. In a previous study, less was known about the prevalence and treatment outcome of FOLFIRINOX of BRCA 1/2 mutation in Asia [14]. The high ORR of 71.4% with FOLFIRINOX therapy in BRCA-positive patients suggests that platinum therapy may be particularly desirable for this subset of patients in clinical scenarios marked by high disease burden, symptomatic disease, and for patients with PDAC. This study may help guide treatment decisions for patients with PDAC.

  1. Is there an objective explanation why their PDAC cohort is younger than other cohorts already described, as mentioned in the discussion section (page 9, line 261-262)?

Reply: Thank you for your good comment. Because germline testing for pancreatic cancer is not covered by insurance in South Korea, we were unable to perform genetic testing on all PDAC patients. Relatively young patients or patients who have a family history of cancer were enrolled.

  1. Can the authors shortly mention, at least in the Materials and Methods section, how the family history of cancer was evaluated: reported cancer types, first-degree relatives?

Reply: Thank you for your careful comment. In the present study, we obtained the family history of cancer from medical records. Of the 20 patients who have family history of cancer, 10 (22.7%) were pancreatic cancer, 3 (6.8%) were breast cancer, and 12 (27.3%) were other malignancy. It was not possible to identify if these individuals were first-degree relative or not. We added this information in the results section.

In Result, page 4, line 153-155

After :

A higher percentage of patients in the BRCA-positive group had prior history of malignancy (6/9, 66.7% vs. 13/55, 23.6%, P=0.016), and breast cancer (4/9, 44.4% vs. 2/55, 3.64%). Of the 20 patients who have family history of cancer, 10 (22.7%) were pancreatic cancer, 3 (6.8%) were breast cancer, and 12 (27.3%) were other malignancy. It was not possible to identify if these individuals were first-degree relative or not. The proportion of family history of any malignancy and number of metastatic sites was greater in the BRCA-positive group than in the BRCA-negative group, however, the difference was not significant (all, P > 0.05).

  1. The authors must describe the meaning of each mentioned abbreviation for each figure and table, in the corresponding legends. The same applies for the manuscript as some abbreviations are not described e.g. NCCN (page 2, line 45). Moreover, to improve the readability of the figure and table legends, can the authors add a comma between the abbreviation and their meaning and the semi-colon between terms, when they list them?

Reply: Thank you for your important comment. According to reviewer’s comments, we corrected the all sentences. And we described the meaning of abbreviation in each figure and table and added a comma and semi-colon properly. We modified this information in the manuscript.

In Introduction, page 2, line 44-46

After :

Pancreatic ductal adenocarcinoma (PDAC) is expected to become the second leading cause of cancer-related deaths in the US before 2030 [1]. In the NCCN (national comprehensive cancer network) guidelines, germline testing is recommended for patients with PDAC using comprehensive gene panels for hereditary cancer syndromes.

  1. The authors must correct the styling of gene names in order to always respect the gene nomenclature recommendations provided by the HGNC (italics for mRNAs and genes, capitals for proteins, etc.) in the manuscript and in tables.

Reply: Thank you for your precious comment. We corrected the styling of gene names in the manuscript and in tables.

  1. The equal contribution of first coauthors looks weirdly mentioned in the first part of manuscript: “These authors contributed equally as second coauthors” (page 1, line 13).

Reply: Thank you for your precious comments. We modified authorship and we deleted “These authors contributed equally as second coauthors” in the first part of manuscript.

In first part of manuscript, page 1, line 5-12

After :

Ji Hoon Park1†, Jung Hyun Jo1, Sung Ill Jang2, Moon Jae Chung1, Jeong Youp Park1, Seung Woo Park1, Seungmin Bang1, Si Young Song1, Hee Seung Lee1*, and Jae Hee Cho2*

1Department of Internal Medicine, Institute of Gastroenterology, Yonsei University College of Medicine, Seoul, Korea
2Division of Gastroenterology, Department of Internal Medicine, Gangnam Severance Hospital, Yonsei University College of Medicine, Seoul, Korea
*Correspondence: lhs6865@yuhs.ac (H.S.L), jhcho9328@yuhs.ac (J.H.C); Tel.: +82-2-2228-1935 (H.S.L), +82-2-2019-3310 (J.H.C)

  1. The reviewer recommends a careful double checking of the spelling mistakes and typos, as numerous ones can be found across the manuscript, e.g. “FOLFIRNOX” (page 1, line 20 and page 2, line 65), “polyp ADP ribose polymerase” (page 2, line 52), “varient” (Supplementary Table 1).

Reply: Thank you for your important comment. We corrected the words. And we performed a careful double check in the manuscript.

After :

“FOLFIRNOX” (page 1, line 20 and page 2, line 65) to “FOLFIRINOX”

“polyp ADP ribose polymerase” (page 2, line 52) to “poly ADP ribose polymerase”

“varient” (Supplementary Table 1) to “variant”

  1. Finally, I would kindly recommend the authors to reconsider their article title, as it could be easily shortened and it mentioned two times “in patients” and “in Korean patients”.

Reply: Thank you for your precious comment. We agree with your opinion, and we deleted ‘in patients’ from the sentence in title.

In Title, page 1, line 2-4

After :

BRCA 1/2 germline mutation predicts the treatment response of FOLFIRINOX with pancreatic ductal adenocarcinoma in Korean patients

Reviewer 3 Report

In this study, Ji Hoon and Dr. Song evaluated the proportion of BRCA 1/2 germline mutations in Korean patients with PDAC and its effect on the chemotherapeutic response of FOLFIRINOX. The authors concluded that patients with BRCA1/2 mutation in the germline blood test had a higher response rate to FOLFIRINOX chemotherapy in PDAC. The authors proposed that the high proportion of BRCA 1/2 germline mutations and response rate supports the need for germline testing to predict better treatment response. Overall, it is an interesting study, and the novelty and strengths of this manuscript are high and fit into the main theme of the journal. There is just one minor issue that should be addressed:

This study included some BRCA1/2 missense mutations (almost 50%) (variant of unknown significant mutations) in their cohort. The authors should discuss it in the discussion section regarding any clinical significance and prognostic value of BRCA1/2 missense mutations in patients with PDAC?  

Author Response

In this study, Ji Hoon and Dr. Song evaluated the proportion of BRCA 1/2 germline mutations in Korean patients with PDAC and its effect on the chemotherapeutic response of FOLFIRINOX. The authors concluded that patients with BRCA1/2 mutation in the germline blood test had a higher response rate to FOLFIRINOX chemotherapy in PDAC. The authors proposed that the high proportion of BRCA 1/2 germline mutations and response rate supports the need for germline testing to predict better treatment response. Overall, it is an interesting study, and the novelty and strengths of this manuscript are high and fit into the main theme of the journal. There is just one minor issue that should be addressed:

This study included some BRCA1/2 missense mutations (almost 50%) (variant of unknown significant mutations) in their cohort. The authors should discuss it in the discussion section regarding any clinical significance and prognostic value of BRCA1/2 missense mutations in patients with PDAC?

Reply: Thank you for your precious comment. We agree with your suggestion, and we inserted the contents about variant of unknown significant mutations in discussion section which you recommended.

In Discussion, page 8, line 229-245

After:

Clinical significance and prognostic value of germline BRCA pathogenic mutation in tumors are well known but whether missense variants of uncertain significance (VUS) have clinical impact is not known. Variants in the gene were often classified as VUSs because of an insufficient understanding of the gene’s role. Variants can be re-classified from VUS to likely pathogenic and further, to pathogenic. Phosphorylation of BRCA1/2 play an important role in their function as regulators of DNA repair, transcription, and cell cycle in response to DNA damage. Tram et al. suggested that VUS has the potential to interfere with the phosphorylation process via abolishing or creating phosphorylation sites on BRCA1/2 [6]. Hu et al. reported that germline VUS variant carriers had superior disease-free survival when compared with wild-type PDAC patients receiving adjuvant chemotherapy (16.5 months vs. 13.1 months, P = 0.007) [7]. Previous statistics indicate that between 10-20% of BRCA sequencing results are VUS, and of these, more than 50% are missense mutations [8]. In this study, BRCA1/2 missense mutations (VUS) were detected in 15.2% in our cohort (supplementary table 1). The ORR was significantly higher in BRCA-positive patients including missense mutations of VUS than in BRCA-negative patients (7/12, 58.3% vs. 3/31, 9.7%, P = 0.002). With further accumulation of data in the future, VUS can be reclassified as pathogenic. We added this information in the Discussion section.

Round 2

Reviewer 1 Report

The authors have added some additional testing but the cohort is still very small. No conclusions for clinical practice can be drawn from such a small cohort. Even for a single country, a larger cohort is required.

Author Response

We were very pleased to know that our manuscript (cancers-1469089) has been subject to opportunity of revision for publication in Cancers. We have carefully considered the valuable comments and suggestions provided by the editor and made great efforts to improve the manuscript accordingly. The followings are point-by-point answers to specific questions raised by reviewer. We hope that the revised version of manuscript could meet the priority required for the publication.

Academic Editor#1 Decision: Accept after minor revision

Notes for Authors: Dear Authors, The paper has been revised according the recommendations of the Reviewers, except for the small size of cohort study (nine BRCA+ patients). However, the manuscript is interesting and this point could be overcome but other points remain to adjust:

We are pleased to hear the editor’s comment for our manuscript, and we also appreciate with the affirmative feedback and comments of the reviewer. We have revised our manuscript considering all the comments. Changes have been made by changing the color to RED in the revised manuscript to avoid any confusion. We hope that we have addressed all the comments and the changes will be considered satisfactory.

  1. Is there a reason to restrict MSI analysis on 13 of all 66 patients? Which is the contribution of this alteration to the treatment response of FOLFIRINOX in pancreatic ductal adenocarcinoma patients?

Reply:

Thank you for your important comment. Of 66 patients, microsatellite instability (MSI) test on tumor tissue was performed in 13 patients. Only two patients (2/13, 15.4%) had a MSI-H. Because genetic testing for pancreatic cancer is not covered by insurance in South Korea, we do not perform MSI testing in all patients. Also, if there was not enough tumor tissue, MSI test could not be performed. The prevalence of MSI in PDAC is very low. Hu et al. reported frequency of MSI in PDAC (7/833, 0.8%) [1]. Furthermore, because this study was a retrospective study, patients who did not undergo MSI test were included.

Of 66 patients, 47 patients of the study participants received FOLFIRINOX chemotherapy. Of these 47 patients, 4 patients did not receive their response evaluation and were excluded from the ORR analysis. Unfortunately, two MSI-H patients did not receive their response evaluation and were excluded from the ORR analysis. We could not analyze the response of FOLFIRIRNOX in MSI-H patients.

  1. Please, introduce MSI analysis in materials and methods.

Reply: Thank you for your valuable comment. We added the detailed information about MSI analysis in Materials and Methods section.

In Materials and Methods, page 3, line 114-118

After :

DNA was extracted from the formalin-fixed, paraffin-embedded blocks of each patient's tumor specimen and amplified using the PCR to investigate the status of microsatellite instability (MSI). Microsatellite instability was identified based on the five instability markers in the Revised Bethesda Guidelines [2]. If the number of instability markers was 2 or more, the tumor was clinically defined as microsatellite instability-high (MSI-H)

  1. Line 245: What means the following phrase “We added this information in the Discussion section” in Discussion section? It is probably wrong.

Reply:

We appreciate the reviewer’s keen comment. This phrase has been deleted because it was wrong.

Reference

  1. Hu, Z.I.; Shia, J.; Stadler, Z.K.; Varghese, A.M.; Capanu, M.; Salo-Mullen, E.; Lowery, M.A.; Diaz, L.A., Jr.; Mandelker, D.; Yu, K.H.; et al. Evaluating Mismatch Repair Deficiency in Pancreatic Adenocarcinoma: Challenges and Recommendations. Clin Cancer Res 2018, 24, 1326-1336, doi:10.1158/1078-0432.Ccr-17-3099.
  2. Umar, A.; Boland, C.R.; Terdiman, J.P.; Syngal, S.; de la Chapelle, A.; Rüschoff, J.; Fishel, R.; Lindor, N.M.; Burgart, L.J.; Hamelin, R.; et al. Revised Bethesda Guidelines for hereditary nonpolyposis colorectal cancer (Lynch syndrome) and microsatellite instability. J Natl Cancer Inst 2004, 96, 261-268, doi:10.1093/jnci/djh034.

Reviewer 2 Report

The authors have satisfactorily responded to my comments and questions, and made the necessary changes to the manuscript and figures.

The manuscript is acceptable for publication in Cancers.

Author Response

(The authors gave the same response as above.)
